# FLORA: Generalizable Motion-Flow-Based Reward Shaping for Scalable Real-World Robot Learning

## Abstract

Rewards design is a long-standing challenge in Reinforcement Learning (RL) for robotics, particularly when scaling to real-world robot tasks. Generally speaking, existing reward design approaches in real-world RL rely either on sparse rewards, which provide little feedback and commonly lead to inefficient learning, or on pre-trained vision-based reward models, which typically lack theoretical guarantees and often fail in generalizing to new tasks. To address these challenges, we introduce **F**low-based **L**anguage-driven **O**ffline **R**eward **A**daptation (**FLORA**), a framework that combines strong generalization capability with a theoretical guarantee of optimal policy invariance. FLORA adopts large language models (LLMs) to automatically generate analytical reward functions for new tasks, leveraging their inherent generalization ability across diverse tasks. Unlike end-to-end neural reward models, these analytical reward functions encode task-relevant priors, enabling efficient few-shot adaptation. With only **3–5** demonstrations, our proposed offline reward improvement procedure optimizes both the structure and parameters of the rewards, producing reliable signals for new tasks. To enable direct operation from raw visual inputs and eliminate the reliance on privileged states, we extract flows from images as inputs to the analytical reward functions. Furthermore, we propose a PBRS-Milestone rewards shaping structure to reformulate rewards signals, which improves practicality while preserving optimal policy invariance guarantee. Extensive experiments show that FLORA enables sample-efficient RL on new tasks, outperforming strong baselines by more than **2×** in simulation, and solving complex real-world manipulation tasks in ∼**20 minutes**, where existing baselines fail even after **60** minutes training. These results establish our method as a critical step towards scalable real-world robot learning.

## 1 Introduction

Designing effective reward functions lies at the heart of reinforcement learning, which remains one of the most stubborn bottlenecks in bringing learning-based control to real-world robots. In RL, the reward design must not only specify what a task is, but also provide signals that are informative, interpretable, and temporally consistent. Such a valid reward design would guide the agent's exploration while preserving policy invariance. In practice, however, rewards in real-world RL are either sparse (Luo et al., 2024; Chen et al., 2025; Luo et al., 2025), offering little feedback and leading to inefficient learning, or hand-crafted (Guzey et al., 2025), requiring significant human effort, domain expertise, and careful tuning, while still being brittle and task-specific.

Recent work has attempted to overcome these challenges. Vision-Language Models (VLMs) (Ma et al., 2024; Rocamonde et al., 2023) provide semantic grounding that is potentially informative for reward design in RL. However, they commonly fail to directly deliver reliable or temporally consistent signals, limiting their direct application in real-world robotics learning. Vision-based reward models (Ma et al., 2023a; Alakuijala et al., 2025) offer dense and reliable feedback signal, but are typically confined to narrow, in-distribution regimes with limited generalization to new tasks. Language-conditioned reward functions (Ma et al., 2023b; Heng et al., 2025) improve interpretability and self-consistency. However, they often rely on privileged state information, which cannot operate directly on visual input. In addition, such paradigms require repeated rounds of costly RL

training in general to improve LLM-generated reward functions before implementation, which is infeasible for real-world RLdeployment. Moreover, none of these approaches guarantee *optimal policy invariance*, raising the risk that policies converge to behaviors misaligned with human intent. These limitations expose a fundamental question: ***Can we design a reward model that is truly generalizable and practical for real-world robot learning?***

To address this question, we propose **F**low-based **L**anguage-driven **O**ffline **R**eward **A**daptation (**FLORA**). Our approach is generalizable, capable of guiding diverse manipulation tasks with only a handful of demonstrations per task. It is reliable, providing interpretable and temporally consistent reward signals that are grounded in task flows while preserving optimal policy invariance. It is also practical, supporting deployment in real-world reinforcement learning without requiring privileged information or expensive end-to-end training of the full RL loop to refine reward functions.

By unifying generalizability, reliability, and practicality, our approach takes a critical step toward scalable and robust real-world robotic reinforcement learning. See website at https://github.com/anonymous-submit-2026/FLORA.

## 2 Related Work

**Traditional Reward Design Methods**    In RL, rewards are often handcrafted by experts to encode task objectives. While this has led to strong results in games and some robotic tasks (Mnih et al., 2015; Silver et al., 2016), it is almost impossible to scale to diverse robotic settings due to the heavy human effort and expertise required for each task. Inverse Reinforcement Learning (IRL) (Arora & Doshi, 2021) aims to reduce this burden by inferring reward functions from expert demonstrations. However, IRL typically needs millions of interactions to learn reliable rewards, making it prohibitively expensive for real-world RL. In contrast, our method generates reliable reward functions automatically without requiring pre-deployment environment interactions.

**VLM-based Reward Design Approaches**    Another line of work leverages Vision-Language Models (VLMs) (Rocamonde et al., 2023; Ma et al., 2024; Kim et al., 2025), either by comparing task descriptions with visual inputs (Rocamonde et al., 2023), prompting VLMs to output reward scores (Ma et al., 2024), or using preferences over image-task pairs (Venkataraman et al., 2024; Ghosh et al., 2025). Thanks to their inherent generalization capacity, VLM-based methods can generalize across many tasks. However, limited robot-specific data in VLM training leads to unstable and temporally inconsistent reward signals, hindering their reliability in real-world RL. By grounding reward functions analytically and refining them with offline robot datasets, our method produces stable, consistent signals well-suited for practical deployment.

**Image-based Reward Models**    To alievate the above-mentioned challenges, Image-based reward shaping methods (Chen et al., 2021; Cui et al., 2022; Fan et al., 2022; Nam et al., 2023; Yang et al., 2024a;b; Kim et al., 2025) proposed to train neural reward models, which can map raw visual inputs directly to scalar signals, on robot dataset. Approaches such as LIV (Ma et al., 2023a) and VLC (Alakuijala et al., 2025) achieve strong performance by producing precise, reliable rewards across several robotic tasks. However, these models generally lack theoretical guarantees and often struggle to generalize beyond training domains. In contrast, our framework leverages LLMs to generate analytical reward functions that embed task-relevant priors and support efficient few-shot adaptation. By reformulating rewards in a PBRS-Milestone structure, we further improve practicality while preserving optimal policy invariance.

**LLM-based Reward Design Approaches**    Large language models (LLMs) (Xie et al., 2023; Ma et al., 2023b; Heng et al., 2025) have demonstrated strong ability to synthesize task-specific reward functions, either by imposing predefined structures or through reflective refinement, achieving human-level design quality in some tasks. However, these approaches often assume access to privileged state information—rarely available in real-world settings—and require repeated RL training cycles to gather performance metrics for rewards improvements, making them prohibitively costly in real-world RL. In contrast, our method introduces a surrogate validation and refinement procedure for LLM-generated code, enabling offline reward learning without repeated full training runs. Crucially, our method also removes the reliance on privileged state information, thereby addressing one of the central obstacles encountered by prior approaches in real-world robotic learning.

## 3 BACKGROUND

Reinforcement learning (RL) can be formulated as a Markov Decision Process (MDP), defined by the tuple $M = (\mathcal{O}, \mathcal{A}, R, \mathcal{T}, \gamma)$. Here, $\mathcal{O}$ denotes the observation space, $\mathcal{A}$ the action space, $\mathcal{R}$ the reward function, $\mathcal{T}$ the transition dynamics, and $\gamma \in (0, 1]$ the discount factor. The agent's objective is to maximize the expected discounted return: $G = \sum_{t=0}^{\infty} \gamma^t R(s_t, a_t)$.

In this work we consider environments with *sparse rewards*, where feedback is provided only upon reaching a goal state. Formally, the reward is defined as

$$R(s, a, s') = \begin{cases} r_s & \text{if } s' = s^g, \\ 0 & \text{otherwise,} \end{cases} \tag{1}$$

where $s^g$ denotes the goal state, $r_s > 0$ is the success reward. The lack of guiding signal makes policy optimization exceptionally challenging.

### 3.1 REWARDS SHAPING

Reward shaping addresses this challenge by augmenting the original reward with additional informative signals. We formalize this by defining a transformed MDP $M'$ as $M' = (\mathcal{O}, \mathcal{A}, \mathcal{R}', \mathcal{T}, \gamma)$, where the new reward is defined as

$$R'(s, a, s') = R(s, a, s') + F(s, a, s'), \tag{2}$$

and $F(s, a, s')$ is a bounded shaping function designed to provide additional dense signals.

### 3.2 POTENTIAL-BASED REWARD SHAPING

Potential-Based Reward Shaping (PBRS) (Ng et al., 1999) is a well-studied approach that encodes heuristic knowledge into the reward function via a potential function $\phi(s)$. Specifically, the shaping term is defined as

$$F(s, a, s') = \gamma \phi(s') - \phi(s), \tag{3}$$

where $\phi : \mathcal{O} \to \mathbb{R}$ assigns each state a scalar "potential."

A crucial theoretical property of PBRS is **policy invariance**: any optimal policy of the shaped MDP $M'$ is also optimal in the original MDP $M$. This allows agents to benefit from denser, more informative signals that accelerate exploration and learning, without altering the final learning objective.

Consequently, PBRS presents a theoretically appealing framework for reward shaping. However, directly applying PBRS into complex, real-world domains like robot learning remains almost infeasible, due to the following reasons:

- PBRS typically assumes access to privileged states (e.g., object poses), which are unavailable in realistic visual robotics settings;
- Manually designing potential functions $\phi$ for diverse tasks is infeasible and does not scale;
- PBRS can be fragile in high-dimensional manipulation: sparse high-potential regions lead to poor exploration and low returns.

**LLMs for Potential Design.** Recent progress in LLMs (Achiam et al., 2023; Liu et al., 2024) suggests a way forward: LLMs can leverage common sense and task knowledge to automatically propose candidate potential functions (Ma et al., 2023b; Heng et al., 2025), reducing the dependence on hand-crafted design. However, this approach introduces new difficulties. LLM-generated potentials are often noisy or inconsistent, and their quality can only be optimized after deployment in reinforcement learning—a process prohibitively costly in the real world.

In summary, while LLMs partially mitigate the challenge of manual potential design, three key obstacles remain: (i) dependence on privileged states, (ii) the fragility in high-dimensional domains, and (iii) the need for efficient refinement and validation of LLM-generated potentials before deployment. Our method FLORA systematically addresses these issues, enabling practical and scalable reward shaping for real-world robot learning.

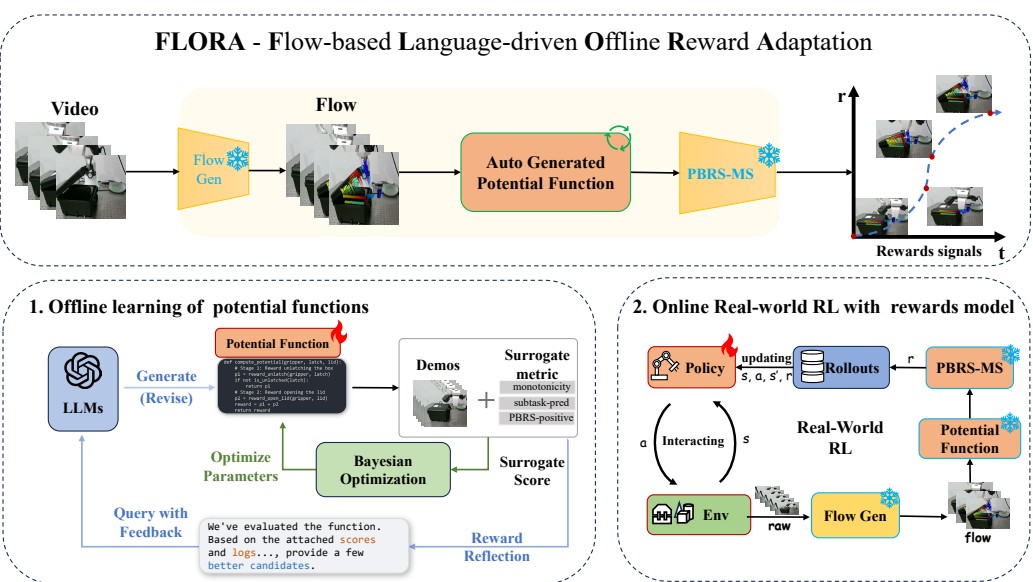

Figure 1: Framework Overview: FLORA consists of a flow generator that extracts motion flows from images, an an auto-generated and auto-optimized potential function potential function that computes potential values from these flows and a PBRS-MS module that produces final reward signals. The framework operates in two stages: (i) Offline learning of potential functions: candidate potential functions are optimized using a surrogate score with Bayesian optimization, while LLMs refine their structure. (ii) Online Real-world RL with rewards model: FLORA can obtain rewards from visual images and then store them into replay buffer to train the policy.

## 4    TOWARDS GENERALIZABLE AND PRACTICAL SHAPING FRAMEWORKS

To overcome the challenges of applying PBRS in real-world robot learning, we propose a shaping framework built on three key components: (i) Flow-based representation for potentials that replace privileged states with relational visual representations; (ii) Surrogate validation and refinement of potentials via pre-validation criteria, LLM self-reflection, and Bayesian optimization; (iii) A robust rewards shaping formulation that preserves policy invariance while improving stability in high-dimensional and imperfect potential function settings. Together, these components form FLORA, a unified framework that is generalizable and practical, enabling real-world reinforcement learning with dense rewards signals.

### 4.1    FLOW BASED REPRESENTATION FOR POTENTIALS

Standard PBRS requires access to privileged states such as object poses—inputs rarely available in realistic robotic setups. Our key insight is that potential functions do not actually depend on precise object positions or orientations; instead, what matters is the *relative spatial relationships* between the gripper and objects, or between objects themselves.

Motivated by this observation, We therefore propose a *flow-based representation* of potentials. Instead of relying on precise state information, we construct object flows that capture relational dynamics directly from raw videos. This approach leverages modern vision models for object localization and point tracking. To improve accuracy, we combined with a light semi-automatic annotation step used once per task to specify relevant regions (e.g., gripper, manipulated objects, target). Full details of the pipeline, including segmentation, transfer matching, and spatio-temporal flow extraction, are provided in Appendix A.

Given object observations $o$ and $o'$ obtained from the flow representation, the shaped reward takes the PBRS form:

$$F(s, a, s') = F(o, a, o') = \gamma\phi(o') - \phi(o), \tag{4}$$

where $\phi(\cdot)$ is a potential function defined over the flow space rather than privileged states. This formulation enables policy-invariant shaping that is grounded in observable motion flow.

## 4.2 Surrogate validation of LLM-generated potentials

Typically, validating and refining LLM-generated potential functions requires multiple rounds of RL training that is prohibitively expensive in real-world robotics (Dulac-Arnold et al., 2019). We instead propose a lightweight surrogate validation framework that improves the rewards function on an offline dataset prior to costly RL training.

**Evaluation dataset.** We collect a small set of successful demonstrations $D_{\text{demo}}$ for the target task. Each video is processed into flow trajectories $\{(o_t, \mu_t)\}_{t=1}^T$, where $o_t$ are object-flow observations and $\mu_t$ are subtask labels. This compact dataset provides supervision for the surrogate evaluation.

**Automatic annotation.** Subtask labels are generated automatically using a GPT4.1 (Achiam et al., 2023). Given task descriptions and demonstration videos, GPT4.1 segments trajectories into subtasks and assigns labels to each frame. To improve consistency, we query GPT4.1 multiple times per frame and adopt the majority label (details in Appendix B). This eliminates the need for human annotations while preserving sufficient reliability.

**Surrogate criteria.** We score each candidate potential function $\phi$ by measuring its alignment with properties of an *ideal* shaping function: (i) correctly predict task stages; (ii) increase monotonically with task progress; (iii) produce positive shaping signals under the PBRS formulation.

The overall surrogate score is

$$\mathcal{J}_s(\phi) = \lambda_1 \, \mathcal{C}_{\text{stage}}(\phi) + \lambda_2 \, \mathcal{C}_{\text{prog}}(\phi) + \lambda_3 \, \mathcal{C}_{\text{pbrs}}(\phi), \tag{5}$$

where $\mathcal{C}_{\text{stage}}$, $\mathcal{C}_{\text{prog}}$ and $\mathcal{C}_{\text{pbrs}}$ are defined below.

**Stage prediction.** Given a demonstration trajectory $(o_t, \mu_t)_{t=1}^T$, an LLM-generated potential is prompted returns both a predicted substage $\hat{\mu}_t$ and a value $\phi_{\hat{\mu}_t}(o_t)$. We require the predicted substage $\hat{\tau}_t$ to align with the subtask label $\mu_t$. This is scored as stage prediction accuracy:

$$\mathcal{C}_{\text{stage}}(\phi) = \frac{1}{T} \sum_{t=1}^T \mathbf{1}\{\hat{\mu}_t = \mu_t\}, \tag{6}$$

where $\mathbf{1}\{\cdot\}$ is the indicator function.

**Progress monotonicity.** Within each trajectory, potentials should grow with progress. Let normalized progress be $p_t = t/T$. We compute the Pearson correlation $\text{Corr}(\cdot, \cdot)$ (Sedgwick, 2012) between the predicted potentials and $p_t$:

$$\mathcal{C}_{\text{prog}}(\phi) = \text{Corr}(\phi_{\hat{\tau}_t}(o_t), \, p_t). \tag{7}$$

A high correlation indicates that the sub-potential values are proportional to progress toward success.

**PBRS Positivity.** Finally, we check that PBRS-shaped signals remain predominantly non-negative, ensuring useful shaping rewards:

$$\mathcal{C}_{\text{pbrs}}(\phi) = \frac{1}{T} \sum_{t=1}^T \mathbf{1}\{F(o_t, a_t, o_{t+1}) \geq 0\}. \tag{8}$$

## 4.3 Hybrid Offline Learning of potential functions

Large language models are effective at proposing commonsense-structured potential functions but often lack precision in assigning continuous parameters. To improve the quality of LLM generated potential functions, we design a hybrid Offline Learning procedure that combines LLM reflection with Bayesian optimization (BO) (Frazier, 2018): the LLM refines the structural form of potential functions, while BO tunes their numerical parameters.

**Bayesian Optimization.** An LLM-generated potential can be written as $\phi(o \mid \theta)$ with tunable parameters $\theta$. The objective of BO is to find parameters that maximize the surrogate score:

$$\theta^\star = \arg\max_\theta \mathcal{S}(\phi; \theta). \tag{9}$$

BO is applied to each LLM-generated function to optimize its parameters. The algorithm is shown in the Appendix C.

**LLM Reflection.** After parameter optimization, we provide the LLM with structured feedback consisting of: (i) candidate potential functions with their optimized parameters, (ii) their corresponding surrogate scores, and (iii) a visualization of potential values evolving over demonstration trajectories. Conditioned on this feedback, the LLM iteratively proposes refined candidates, progressively improving functional structure.

The hybrid Refinement Procedure is shown in the Algorithm 1.

### 4.4 A ROBUST REWARDS SHAPING FORMULATION

After the hybrid offline learning of potential functions, we obtain a potential function that is acceptable but still maybe not perfect. Directly applying such a function in high-dimensional reinforcement learning can lead to training instabilities, particularly for long-horizon tasks. We identify two main causes of this issue. For the following theoretical analysis, we revert to the notation $\phi(s)$ to maintain consistency with the classical PBRS framework.

**(i) Limited exploration due to imperfect potential functions.** Although refinement enforces potentials that are generally monotonic with task progress, the resulting function may still contain small local minima or maxima, such as e.g., $\phi(s_{t+1}) - \phi(s_t) \approx 0$ in a local basin. In such regions the agent receives little or misleading shaped reward, and exploration stagnates. The effect is compounded in long-horizon tasks: even a few local irregularities ("bumps" or "ridges" in $\phi$) can repeatedly trap the agent in suboptimal regions, preventing successful rollouts.

**(ii) Sparsity and fragility of high-potential regions.** Let $\mathcal{S}_{\text{high}} = \{s \mid \phi(s) \geq \tau\}$ denote states with high potential. In high-dimensional tasks, $\frac{\mu(\mathcal{S}_{\text{high}})}{\mu(\mathcal{S})} \ll 1$, where $\mu$ is the Lebesgue measure over states. Thus, high potential regions are quite sparse. Moreover, $\phi$ often exhibits sharp discontinuities: two visually similar states $s$ (object grasped securely) and $s'$ (object just slipped) may yield $|\phi(s) - \phi(s')| \gg 0$, even though $\|s - s'\|$ is small. This discontinuity causes sharp drops in return; once an agent falls from $\mathcal{S}_{\text{high}}$ into a low-potential state, the cumulative return $R = \sum_t r_t'$ may collapse, regardless of previously achieved progress. Because RL credit assignment spreads rewards across many steps, the Q-function $Q(s, a)$ may learn to undervalue the entire region near $s$. Hence, the Q-function fails to separate promising from unpromising states, leading to instability during training.

To address this challenge, we propose **PBRS-Milestone** (PBRS-MS), a new reward shaping method that augments PBRS with global signals at predefined subgoals. Instead of relying solely on local potential differences, we introduce *milestone rewards*: large positive signals delivered when the agent first crosses certain progress thresholds.

**Milestone Rewards.** Let $\phi : \mathcal{S} \to \mathbb{R}_{\geq 0}$ be the potential function, and let $\phi_{\max} = \max_{s \in \mathcal{S}} \phi(s)$ denote its maximum. We define a sequence of milestone thresholds

$$\mathcal{K} = \{\kappa_1, \kappa_2, \ldots, \kappa_K\}, \quad \text{with } \kappa_i \in (0, 1), \ \kappa_1 < \kappa_2 < \cdots < \kappa_K. \tag{10}$$

Each threshold corresponds to a *milestone set*

$$\mathcal{M}_i = \{s \in \mathcal{S} \mid \phi(s) \geq \kappa_i \, \phi_{\max}\}, \quad i = 1, \ldots, K. \tag{11}$$

Let $\mathcal{S}$ be the original state space. We define an augmented state space $\tilde{\mathcal{S}} = \mathcal{S} \times \{0, 1, \ldots, K\}$, where $\tilde{s} = (s, m)$ consists of the original state $s$ and the current milestone index $m$. The milestone index $m$ records the highest milestone that has been achieved up to the current time, i.e.,

$$m_t = \max\{i \mid \exists \, t' \leq t : s_{t'} \in \mathcal{M}_i\}.$$

The milestone index updates deterministically as $m_{t+1} = \max(m_t, \max\{i \mid s_{t+1} \in \mathcal{M}_i\})$.

Instead of a one-time bonus, the milestone achievement is incorporated through a separate potential-based shaping term. We define a *milestone potential function* $\Psi : \{0, 1, \ldots, K\} \rightarrow \mathbb{R}$ on the milestone index space:

$$\Psi(m) = \sum_{i=1}^{m} R_i,$$

where $R_i > 0$ is the bonus value associated with milestone $i$. This function $\Psi(m)$ quantifies the total cumulative bonus for achieving up to the $m$-th milestone.

**PBRS-Milestone Reward Shaping.** The final shaped reward in the augmented MDP is now defined as the sum of the original reward, the local potential-based shaping from $\phi$, and a **global milestone-based shaping** from $\Psi$:

$$r'_t = r_t + \underbrace{\gamma\phi(s_{t+1}) - \phi(s_t)}_{\text{local PBRS}} + \underbrace{\gamma\Psi(m_{t+1}) - \Psi(m_t)}_{\text{global milestone PBRS}}. \tag{12}$$

**Theorem 1** (Policy Invariance of PBRS-MileStone)**.** *Consider the original MDP* $M = (\mathcal{S}, \mathcal{A}, P, \gamma, R)$ *and the augmented MDP* $\tilde{M} = (\tilde{\mathcal{S}}, \mathcal{A}, \tilde{P}, \gamma, \tilde{R})$ *constructed by the Revised PBRS-Milestone method, with the total reward defined as in eq.* (12)*:*

$$\tilde{R}(\tilde{s}_t, a_t, \tilde{s}_{t+1}) = R(s_t, a_t, s_{t+1}) + (\gamma\phi(s_{t+1}) - \phi(s_t)) + (\gamma\Psi(m_{t+1}) - \Psi(m_t)). \tag{13}$$

*Then, for any discount factor* $\gamma \in [0, 1)$*, the set of optimal policies is invariant between* $M$ *and* $\tilde{M}$*. Specifically, a policy* $\pi^*$ *is optimal in* $M$ *if and only if its natural extension* $\tilde{\pi}^*((s, m)) = \pi^*(s)$ *is optimal in* $\tilde{M}$*.*

*Proof.* The proof is provided in Appendix D. $\square$

## 5 EXPERIMENTS

To evaluate FLORA as a reward learning method for new robot tasks, we conducted a series of experiments on simulated and real-world environments. We mainly focus on the following three questions:

- How well does FLORA generalize across diverse tasks compared to strong baselines?
- Does FLORA perform reliably on Real World Robotic Tasks?
- Which components of FLORA contribute most to overall performance?

### 5.1 SIMULATION EXPERIMENTS

To assess FLORA's generalizability, we evaluate it on eight Meta-World (Yu et al., 2020) tasks spanning diverse motion primitives and task horizons. These tasks cover a wide range of manipulation skills—push, pull, insert, turn, pick, and place—and include both single-stage tasks and long-horizon tasks that require precise, contact-rich interactions. Detailed task descriptions and settings are provided in Appendix E.1.

We compare against several SOTA reward models:(i) **LIV** (Ma et al., 2023a), trained on large-scale real-robot datasets; (ii) **VLC** (Alakuijala et al., 2025), a transferable language-conditioned reward model; (iii) **Dense Rewards**, Meta-World's shaped reward functions with amplified success signals; and (iv) **Sparse Rewards**, the default binary success signals. All methods operate under standard implementations are image- and language-conditioned wherever applicable, and are trained from scratch using a **RLPD** (Ball et al., 2023) agent. The hyberparameters of RLPD are provided in the Appendix E.2.

Figure 2 reports learning curves of RLPD agents trained with these reward functions across eight tasks. As expected, sparse rewards lead to near-zero success rates in most cases, except for relatively simple tasks such as *Drawer Close* and *Hammer*, highlighting the difficulty of exploration in Visual RL without shaped guidance. In contrast, our approach consistently achieves strong and stable performance across all tasks, matching or surpassing Dense Rewards in both sample efficiency and

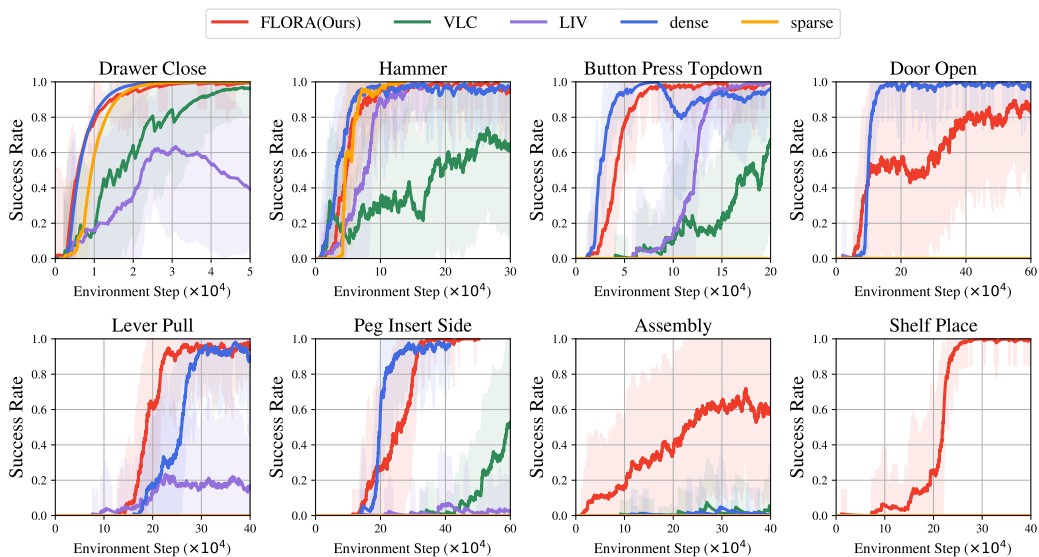

Figure 2: Learning curves of RLPD agents with different reward functions on eight Meta-World manipulation tasks, reported as success rate. Solid lines show the mean, and shaded areas indicate stratified bootstrap intervals over 3 runs.

final success rate. In contrast, FLORA consistently achieves strong performance across all tasks, matching or surpassing Dense Rewards in both sample efficiency and final success rate.

Specifically, FLORA shows rapid convergence on contact-rich tasks like *Lever Pull* and *Peg Insert Side*, where other learning-based reward models such as LIV and VLC struggle to provide sufficiently shaped signals. In long-horizon tasks such as *Shelf Place* and *Assembly*, FLORA delivers significantly higher success rates by guiding RLPD agents with stage-aware potentials, while baselines plateau at suboptimal success rates. Importantly, FLORA requires no manually designed dense rewards, yet closely tracks or even outperforms them, highlighting the effectiveness of our automatic rewards refinement pipeline.

Taken together, these results show two key advantages of FLORA: (i) **Generality** — FLORA works reliably across diverse manipulation tasks with different horizons and contact dynamics; (ii) **Efficiency** — it provides dense progress-aware signals that accelerate training.

## 5.2 REAL-WORLD EXPERIMENTS

To evaluate the practicality of FLORA in realistic settings, we conduct two real-world manipulation tasks that require long-horizon control and precise, contact-rich interactions.

We use a 7-DOF Franka Emika (Haddadin et al., 2022) equipped with wrist- and third-person RGB-D cameras. Policies are conditioned on visual observations and proprioception. Additional implementation details are provided in Appendix F.1.

We have designed two challenging manipulation tasks: (i) **Peg Insert Deep**: aligning and inserting a peg into a tightly fitting hole, demanding sustained contact and millimeter-level precision. (ii) **Box Open**: grasping and rotating a hinged lid, requiring stable grasping and controlled rotational motions. Both tasks demand fine-grained control under physical constraints.

For each task, we collected five teleoperated demonstrations to seed our reward learning pipeline. We train policies using RLPD, a data-efficient visual reinforcement learning algorithm, keeping all hyperparameters fixed for fair comparison. Demonstrations were also included in the replay buffer for faster learning. The hyberparameters are provided in the Appendix F.2.

The experiment results in show in the Table 1. Within approximately 20 minutes of real-world training, FLORA enables RLPD to achieve a nearly 100% success rate. By contrast, SERL (Luo et al.,

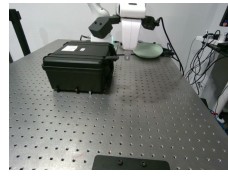

| Task | Ours | SERL |
| | @ 20 min | @ 60 min |
| --- | --- | --- |
| Franka Peg Insert Deep | **20/20** | 0/20 |
| Franka Box Open | **20/20** | 0/20 |

| (a) Peg Insert Deep | (b) Box Open | (c) Real-world experiments results. |

Table 1: Real-world RL Experiments: (a) Franka Peg Insert Task, (b) Franka Box Open Task, (c) Real-world experiments results: the success rate of RLPD agent with different rewards models.

2024) with its default binary classifier-based reward model fails to solve either task even after 60 minutes of training. Failure cases of the baseline highlight the difficulty of these tasks: in *Peg Insert Deep*, policies approach the hole but consistently fail insertion due to the tight clearance tolerance; in *Box Open*, unstable grasps prevent consistent lid rotation. In contrast, FLORA provides dense, stage-aware rewards that guide the policy through critical sub-skills—securing a grasp, stabilizing contact, and executing precise manipulations—leading to rapid and reliable convergence.

Overall, these real-world experiments demonstrate that FLORA produces reliable reward signals in contact-rich manipulation tasks, substantially extends the applicability of SERL to more complex robotic skills, and enables more data-efficient learning. Our results highlight a step forward toward scalable real-world robotic reinforcement learning.

### 5.3 ABLATION STUDY

**Ablation on rewards shaping structure** we first conduct an ablation study with rewards shaping structure, keep all other design same to our method: (1) **PBRS-MS**: the proposed rewards shaping structure in our method; (2) **PBRS Only**: the classical potential-based reward shaping method; (3) **Direct Rewards**, directly using potential values as rewards.

Ablation experiments are performed on the *Lever Pull* and *Peg Insert Side* tasks, with results summarized in Figure 3. We observe that the classical PBRS performs poorly in these high-dimensional settings: agents rarely achieve successful rollouts. The Direct Rewards variant yields higher success but suffers from poor data efficiency. And in some cases, fails to converge to a nearly 100% success rate. This occurs because directly using potential function values may alter the optimality of policies: the optimal policy in the transformed MDP may no longer cor-

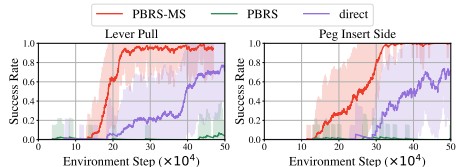

Figure 3: Ablation study on rewards shaping structure.

respond to the optimal policy in the original MDP. In contrast, PBRS-MS demonstrates consistent fast convergence and stable performance, achieving high success rates and highlighting the methodological advantages of our approach.

We also have conducted **ablation study on offline rewards improvement procedure**, whose result is shown in the Appendix G.

## 6 CONCLUSION

In this work, we have introduced FLORA, a motion flow-based reward shaping framework that is both generalizable and practical for real-world robot learning. Our approach automatically constructs reliable, dense reward functions for new tasks while rigorously preserving optimal policy invariance. By operating directly from raw visual inputs without relying on privileged states, our framework enables seamless deployment in real-world RL scenarios. We believe this method substantially reduces the burden of reward engineering and paves the way for scaling reinforcement learning to a wide range of real-world robotic applications.

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

## A  FLOW GENERATION

A natural approach to constructing object flow is to apply open-world object detection models (e.g., DINO-X (Ren et al., 2024), YOLO-World (Cheng et al., 2024)) to localize the gripper and task-relevant objects, followed by point tracking within these regions, just likes Im2flow2act (Xu et al., 2024). However, our experiments show that even state-of-the-art detectors are struggle in our setting: they frequently fail to detect robot grippers, misclassify objects, or produce bounding boxes that are oversized. These errors are particularly common for objects that appear infrequently in their training datasets.

To address this limitation, we design a semi-automatic pipeline for robust flow construction. The pipeline requires minimal human input (only once per task) and improves region selection accuracy.

We construct object flow in two stages, as shown in the Figure 4:

**Region annotation** (one-time per task). We apply SegmentAnything-v2 (Ravi et al., 2024) to a single reference image to obtain a segmentation map. A human annotator selects the relevant regions (e.g., gripper, manipulated objects, targets). We uniformly sample points within these regions, producing the initial flow representation $O_0 \in \mathbb{R}^{3 \times n}$.

**Flow generation on new videos.** For each video, we extract DINO-v2 (Oquab et al., 2024) embeddings from initial frames. Points in $O_0$ are matched to nearest features, yielding a transferred set $O_0' \in \mathbb{R}^{3 \times n}$. Direct correspondence may yield inaccurate matches (e.g., near but not on the target object); we therefore refine them by identifying the segmentation region containing the majority of matches and reinitializing $O_0$ from this region. Finally, we apply TAPIP3D (Zhang et al., 2025) to generate the spatiotemporal flow sequence $O_T \in \mathbb{R}^{3 \times T \times n}$, where $T$ denotes the sequence length.

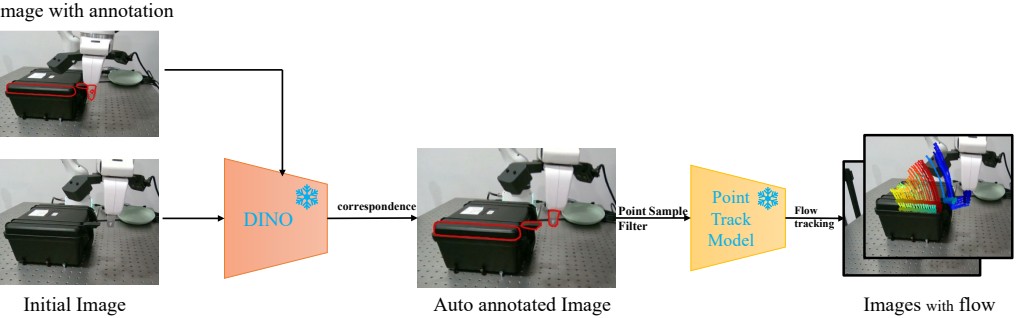

Figure 4: The flow generation procedure.

## B  SUBTASK LABEL

**Automatic annotation of subtask labels.**   Instead of relying on human annotators to label subtasks in $D_{\text{demo}}$, we leverage GPT4.1 for automatic annotation. Specifically, we first provide the VLM with a textual description of the task and a demonstration video, prompting it to segment the task into a sequence of subtasks. Given the segmentation criteria, the VLM is queried to assign a subtask label to each frame in the dataset. For each query, we present the VLM with a randomly sampled subset of frames from the same video, arranged in random order, which encourages more consistent predictions across the trajectory. To improve reliability, we query the GPT4.1 multiple times per image and adopt the most frequent label. These automatically generated labels enable surrogate evaluation without human annotation.

## C  HYBRID OFFLINE LEARNING ALGORITHM

The hybrid offline learning algorithm is shown in the Algorithm 1. And the Bayesian Optimization (BO) for tuning parameters $\theta$ is shown in the Algorithm 2.

---

**Algorithm 1** Hybrid Offline Learning Procedure

---

1: **Input:** Task description $l$, environment code $M$, coding LLM $\mathcal{LLM}$,
2:     surrogate score function $\mathcal{S}$, initial prompt $p_0$, Bayesian optimization routine BO
3: **Hyperparameters:** Number of iterations $N$, batch size $K$
4: Initialize prompt: $p \leftarrow p_0$
5: **for** iteration $i = 1, 2, \ldots, N$ **do**
6:     // Generate $K$ candidate reward functions via LLM
7:     $\{(\phi_1, \theta_1), \ldots, (\phi_K, \theta_K)\} \sim \mathcal{LLM}(l, M, p)$
8:     Initialize best score: $s_{\text{best}} \leftarrow -\infty$
9:     **for** each candidate $(\phi_k, \theta_k)$ where $k = 1, \ldots, K$ **do**
10:         // Optimize parameters using Bayesian optimization
11:         $\theta_k^* \leftarrow \text{BO}(\phi_k, \theta_k, \mathcal{S})$
12:         $s_k^* \leftarrow \mathcal{S}(\phi_k; \theta_k^*)$
13:         **if** $s_k^* > s_{\text{best}}$ **then**
14:             $s_{\text{best}} \leftarrow s_k^*$
15:             $\phi_{\text{best}} \leftarrow \phi_k, \theta_{\text{best}}^* \leftarrow \theta_k^*$
16:         **end if**
17:     **end for**
18:     // Update prompt with top-performing candidates
19:     $p \leftarrow p \oplus \text{Reflection}(\phi_{\text{best}}, \theta_{\text{best}}^*, s_{\text{best}})$
20: **end for**
21: **Return:** Optimal reward function $\phi_{\text{best}}$ and parameters $\theta_{\text{best}}^*$

---

**Algorithm 2** Bayesian Optimization (BO) for tuning parameters $\theta$

---

**Input:** Objective function $S(\phi; \theta)$, search space $\Theta$, budget $T$
Initialize dataset $\mathcal{D}_0 = \{\}$
**for** $t = 1, 2, \ldots, T$ **do**
    Fit surrogate model $\hat{S}_t$ using $\mathcal{D}_{t-1}$
    Define acquisition function $a_t(\theta \mid \hat{S}_t)$ (e.g., Expected Improvement)
    Select next query point:
$$\theta_t = \arg\max_{\theta \in \Theta} a_t(\theta \mid \hat{S}_t)$$

    Evaluate true objective: $y_t = S(\phi; \theta_t)$
    Augment dataset: $\mathcal{D}_t = \mathcal{D}_{t-1} \cup \{(\theta_t, y_t)\}$
**end for**
**Output:** Best parameter $\theta^* = \arg\max_{(\theta, y) \in \mathcal{D}_T} y$

---

# D  PROOF OF THEOREM 1

*Proof.* The foundation of this proof is the well-established theorem that potential-based reward shaping (PBRS) preserves optimal policies (Ng et al., 1999). The shaping function must be of the form:
$$F(s, a, s') = \gamma\Phi(s') - \Phi(s), \tag{14}$$
where $\Phi$ is a potential function defined on the state space.

In our Revised PBRS-Milestone method, the total reward in the augmented MDP $\tilde{M}$ is given by:
$$\tilde{R}(\tilde{s}_t, a_t, \tilde{s}_{t+1}) = R(s_t, a_t, s_{t+1}) + F_{\text{local}} + F_{\text{global}},$$
where
$$F_{\text{local}} = \gamma\phi(s_{t+1}) - \phi(s_t), \quad F_{\text{global}} = \gamma\Psi(m_{t+1}) - \Psi(m_t).$$

We can combine these two shaping terms into a single, composite potential-based shaping function defined on the *augmented state space* $\tilde{\mathcal{S}}$. Let us define a new potential function $\Phi_{\text{total}} : \tilde{\mathcal{S}} \rightarrow \mathbb{R}$ as the sum of the local and global potentials:
$$\Phi_{\text{total}}(s, m) = \phi(s) + \Psi(m).$$

Now, consider the shaping function generated by $\Phi_{\text{total}}$:

$$
\begin{aligned}
F_{\text{total}}(\tilde{s}_t, a_t, \tilde{s}_{t+1}) &= \gamma \Phi_{\text{total}}(\tilde{s}_{t+1}) - \Phi_{\text{total}}(\tilde{s}_t) \\
&= \gamma \left( \phi(s_{t+1}) + \Psi(m_{t+1}) \right) - \left( \phi(s_t) + \Psi(m_t) \right) \\
&= \left( \gamma \phi(s_{t+1}) - \phi(s_t) \right) + \left( \gamma \Psi(m_{t+1}) - \Psi(m_t) \right) \\
&= F_{\text{local}} + F_{\text{global}}.
\end{aligned}
$$

This demonstrates that the combined shaping reward $F_{\text{local}} + F_{\text{global}}$ used in our method is indeed a valid potential-based shaping function of the form eq. (14) for the augmented MDP $\tilde{M}$, with the potential function $\Phi_{\text{total}}$ defined on the augmented state space.

Therefore, by the direct application of the policy invariance theorem (Ng et al., 1999), the sets of optimal policies for the original MDP $M$ and the augmented MDP $\tilde{M}$ are identical. The optimal action-value functions $Q^*$ and $\tilde{Q}^*$ are related by:

$$
\tilde{Q}^*((s,m), a) = Q^*(s,a) - \Phi_{\text{total}}(s,m) = Q^*(s,a) - (\phi(s) + \Psi(m)).
$$

Since the subtracted term $\Phi_{\text{total}}(s,m)$ is independent of the action $a$, the action that maximizes $Q^*(s,a)$ is identical to the action that maximizes $\tilde{Q}^*((s,m), a)$ for any augmented state $(s,m)$. This completes the proof. $\qquad\square$

# E    SIMULATION EXPERIMENTS

## E.1    TASKS

We benchmark our method on eight manipulation tasks from the Meta-World benchmark (Yu et al., 2020), chosen to cover diverse motion primitives, object interactions, and task horizons:

- **Drawer Close:** The agent must push a partially opened drawer until it is fully closed. This tests precise pushing control and position tracking.

- **Hammer:** The agent grasps a hammer and uses it to pound a nail into a fixed board. This requires both tool use and coordinated arm control.

- **Button Press Topdown:** The agent presses a button from above until it is fully depressed. This emphasizes vertical positioning accuracy.

- **Door Open:** The agent pulls a door handle and opens the door past a target angle. This requires learning pull motions and hinge dynamics.

- **Lever Pull:** The agent pulls down a lever until it reaches a target threshold. This again combines reaching, grasping, and pulling actions.

- **Peg Insert Side:** The agent aligns a peg with a side-mounted hole and inserts it. This is a long-horizon contact-rich task requiring fine alignment.

- **Assembly:** The agent picks up a nut and places it onto a peg fixed on the table. This requires precise pick-and-place coordination and object handling.

- **Shelf Place:** The agent grasps a block and places it onto a shelf at a target position. This tests long-horizon reaching and placing motions with height control.

These tasks span primitive skills including push, pull, press, open, insert, and place, and together represent a wide spectrum of contact-rich manipulation challenges.

For each task, we collect five demonstration videos using a scripted policy, randomly selecting three for training and two for evaluation. For each video, we extract motion flows with our pipeline and use GPT4.1 to automatically annotate sub-task labels and produce sub-task segmentation criteria. We then provide the task description, environment code, and the generated segmentation criteria to GPT4.1 to generate multiple candidate potential functions. Finally, we apply our offline reward improvement procedure to validate and iteratively optimize these candidates, selecting the one with the highest surrogate score as the final potential.

| Hyperparameter | Value |
|---|---|
| **Policy** | |
| Tanh squash distribution | True |
| Std parameterization | exp |
| Std min | $1 \times 10^{-5}$ |
| Std max | 5 |
| **Critic Network** | |
| Activations | $\tanh$ |
| Use layer norm | True |
| Hidden dims | [256, 256] |
| **Policy Network** | |
| Activations | $\tanh$ |
| Use layer norm | True |
| Hidden dims | [256, 256] |
| **Training** | |
| Temperature init | $1 \times 10^{-2}$ |
| Discount ($\gamma$) | 0.99 |
| Backup entropy | False |
| Critic ensemble size | 4 |
| Critic subsample size | 2 |
| Critic–Actor update ratio | 2 |
| **Optimizers** | |
| Temperature LR | $1 \times 10^{-4}$ |
| Temperature grad clip norm | 1 |
| Actor LR | $3 \times 10^{-4}$ |
| Actor grad clip norm | 1 |
| Critic LR | $3 \times 10^{-4}$ |
| Critic grad clip norm | 1 |

Table 2: Hyperparameters of RLPD used in our simulation experiments.

### E.2 HYPERPARAMETERS

The hyperparameters used for training the RLPD agent are summarized in Table 2.

## F REAL-WORLD EXPERIMENTS

### F.1 ROBOT SETUP

We use a 7-DOF Franka Emika Panda, controlled at 10 Hz in end-effector space. Visual observations come from two RGB-D cameras: an Intel RealSense D405 mounted on the wrist and an Intel RealSense D455 providing a third-person view. The policy receives both visual inputs and robot proprioception.To ensure safe interaction, we employ impedance control as the low-level controller to regulate contact forces and prevent potential damage.

### F.2 HYPERPARAMETERS

The hyperparameters used for training the RLPD agent in the real-world RL tasks are summarized in Table 3.

## G ABLATION STUDY

**Ablation on offline rewards improvement procedure**    we conduct an ablation study with offline rewards improvement procedure, keep all other design same to our method: (1) **LLM Reflection +**

| Hyperparameter | Value |
|---|---|
| **Policy** | |
| Tanh squash distribution | True |
| Std parameterization | exp |
| Std min | $1 \times 10^{-5}$ |
| Std max | 5 |
| **Critic Network** | |
| Activations | $\tanh$ |
| Use layer norm | True |
| Hidden dims | [256, 256] |
| **Policy Network** | |
| Activations | $\tanh$ |
| Use layer norm | True |
| Hidden dims | [256, 256] |
| **Training** | |
| Temperature init | $1 \times 10^{-2}$ |
| Discount ($\gamma$) | 0.96 |
| Backup entropy | False |
| Critic ensemble size | 10 |
| Critic subsample size | 2 |
| Critic–Actor update ratio | 4 |
| **Optimizers** | |
| Temperature LR | $1 \times 10^{-4}$ |
| Temperature grad clip norm | 1 |
| Actor LR | $3 \times 10^{-4}$ |
| Actor grad clip norm | 1 |
| Critic LR | $3 \times 10^{-4}$ |
| Critic grad clip norm | 1 |
| **Replay buffer** | |
| Demos | 5 |

Table 3: Hyperparameters of RLPD used in our real-world RL experiments.

**BO**: our proposed procedure that combines LLM reflection with Bayesian Optimization; (2) **LLM Reflection only**: using LLM reflection alone to refine the reward functions; (3) **BO**: using Bayesian Optimization alone to optimize the reward functions. (4) **Direct**: directly selecting the best LLM-generated reward function from candidates without additional optimization.

For each variant, we repeat the optimization process five times and evaluate using the three proposed metrics: Stage prediction score $\mathcal{C}_{\text{stage}}(\phi)$, Progress monotonicity score $\mathcal{C}_{\text{prog}}(\phi)$ and PBRS positive score $\mathcal{C}_{\text{pbrs}}(\phi)$. Results are reported in Table 4.

| Env | Variants | $\mathcal{C}_{\text{prog}}$ | $\mathcal{C}_{\text{stage}}$ | $\mathcal{C}_{\text{PBRS}}$ | $\mathcal{J}_s$ |
|---|---|---|---|---|---|
| Lever-Pull | LLM Reflection + BO | 0.92±0.01 | **0.90±0.00** | **0.94±0.00** | **9.29±0.04** |
| | LLM Reflection only | 0.82±0.21 | 0.68±0.17 | 0.67±0.14 | 7.31±1.58 |
| | BO only | **0.96±0.01** | 0.78±0.08 | 0.91±0.02 | 9.14±0.03 |
| | Direct | 0.44±0.06 | 0.44±0.00 | 0.58±0.04 | 5.10±0.06 |
| Peg-Insert | LLM Reflection + BO | **0.90±0.05** | **0.88±0.01** | **0.89±0.07** | **8.95±0.15** |
| | LLM Reflection only | 0.85±0.07 | 0.66±0.01 | 0.73±0.10 | 7.69±0.26 |
| | BO only | 0.89±0.06 | 0.75±0.09 | 0.88±0.08 | 8.74±0.06 |
| | Direct | 0.84±0.01 | 0.67±0.01 | 0.58±0.09 | 6.89±0.47 |

Table 4: Ablation study on offline rewards improvement procedure. The evaluation metrics are our proposed surrogate scores: Stage prediction score $\mathcal{C}_{\text{stage}}(\phi)$, Progress monotonicity score $\mathcal{C}_{\text{prog}}(\phi)$, PBRS positive score $\mathcal{C}_{\text{pbrs}}(\phi)$ and total surrogate score $\mathcal{J}_s$
.

As shown in Table 4, the low surrogate score of the *Direct* method indicates that LLM-generated potential functions are not of sufficient quality for direct use. Incorporating *LLM Reflection* improves the quality of these functions, but the resulting rewards still fail to provide signals that are reliably monotonic with task progress. In contrast, *BO* substantially enhances monotonicity, yet its effectiveness is fundamentally limited—it can only tune the parameters of a fixed structure without introducing new semantic insights. Taken together, these findings demonstrate that combining *LLM Reflection* with *BO* achieves the best results, as reflection contributes semantic structure while *BO* enforces quantitative consistency.

## THE USE OF LARGE LANGUAGE MODELS (LLMS)

Large Language Models (LLMs) serve two purposes in our study:

- **Reward function generation.** We employ LLMs to automatically propose candidate reward functions that are subsequently refined through reward shaping. In this capacity, the LLM is an integral component of our algorithmic pipeline.
- **Writing assistance.** We also use LLMs to improve the clarity and consistency of the text, including task descriptions and experimental documentation. This usage is restricted to language refinement and does not affect technical content.

All experimental methods, architectures, and results are solely determined by our proposed algorithms and evaluations, ensuring that LLM usage does not influence decision-making or outcomes.

