# OpenReview forum: "FLORA: Generalizable Motion-Flow-Based Reward Shaping for Scalable Real-World Robot Learning"
_ICLR.cc/2026/Conference — ICLR 2026 Conference Withdrawn Submission_

### Official Review · Reviewer_5Has · 2025-10-24

**Soundness:** 2
**Presentation:** 2
**Contribution:** 2
**Rating:** 2
**Confidence:** 4

**Summary:**

This paper proposes FLORA (Flow-based Language-driven Offline Reward Adaptation), a pipeline for generating and refining potential-based shaping rewards that operate directly from visual inputs. Key components are: (1) a flow-based visual representation that replaces privileged state inputs with object/point motion flows extracted from video; (2) LLM-generated analytical potential functions that are iteratively refined via a hybrid offline procedure combining LLM “reflection” and Bayesian optimization (BO) on a small set of demonstrations using surrogate metrics (stage prediction, progress monotonicity, PBRS positivity); and (3) PBRS-Milestone (PBRS-MS), an augmented potential-based shaping scheme that adds milestone bonuses while preserving policy invariance. Experiments claim that FLORA generalizes across 8 Meta-World tasks, dramatically improves sample efficiency in simulation (matching or beating dense engineered rewards), and solves two challenging real-world Franka tasks within ~20 minutes where a baseline fails.

**Strengths:**

•	Tackles an important practical bottleneck in robot RL: usable dense rewards from vision without privileged states. The idea of grounding PBRS potentials in motion flows is a reasonable and practical substitution for privileged state inputs.

•	Proposes a concrete hybrid offline refinement loop (LLM reflection + BO) and defines clear surrogate metrics (stage accuracy, monotonicity, PBRS positivity) to evaluate candidate potentials prior to RL deployment—this is useful engineering.

**Weaknesses:**

•	The flow extraction requires a “one-time per task” human region annotation. The paper does not quantify the human time/effort, nor compare to fully automatic alternatives; this weakens claims of “scalable” automation.

•	Heavy reliance on LLM-produced labels/structures: surrogate scoring and LLM reflection use GPT4.1 to segment subtasks and to propose potentials. The paper gives few diagnostics on LLM error modes, prompt sensitivity, or how often LLM suggestions need manual correction. If LLM segmentation is noisy, surrogate optimization and BO may optimize toward brittle potentials.

•	many results report 3 runs or single-shot real-robot trials (20/20 claims) without variance estimates or significance testing. Small sample sizes make it hard to assess reproducibility and robustness.

•	Missing ablations on crucial axes: sensitivity to flow quality (e.g., replace TAPIP3D with simpler optical flow), sensitivity to number of demonstrations (they use 3–5 demos but no curve), BO budget and acquisition choice, LLM prompt variants, and the effect of the “semi-automatic” annotation step. These ablations are necessary to show the method’s real-world practicality.

•	Experiments focus on mid-short horizon manipulations (Meta-World tasks and two Franka tasks). It is unclear whether FLORA scales to long-horizon, highly stochastic, or multi-stage tasks where flows may be ambiguous or where milestone thresholds are harder to define.

•	The results in Figure 2 do not seem to strongly prove the advancedness of the proposed method. The performance comparison of the eight tasks is very slight compared with the best baselines, except for the assembly and shelf placement tasks.

**Questions:**

1.	Please quantify the “one-time per task” annotation cost for flow region selection (time per task, number of clicks/frames). Is the pipeline usable when a human cannot or will not annotate regions?

2.	What is the accuracy/consistency of GPT4.1 subtask segmentation on the demo set? Provide confusion matrices or per-frame agreement statistics (LLM vs human) and report how surrogate scores change if the automatic labels are noisy.

3.	What BO budget and acquisition function were used? How many candidate potential structures were evaluated per iteration? Provide wall-clock costs for the offline refinement stage to assess practicality.

4.	Please add ablations varying (a) number of demonstrations used for surrogate evaluation (1→10), (b) flow extractor quality (TAPIP3D vs simple optical flow), (c) BO budget and initialization, and (d) LLM prompt variants. These will clarify whether FLORA’s gains depend on fragile engineering choices.

5.	For the Franka experiments, report variance across multiple independent runs (≥5), show representative failure rollouts for both FLORA and the baseline, and explain any cases where FLORA’s potential misleads the policy.

---

### Official Review · Reviewer_keDw · 2025-10-25

**Soundness:** 2
**Presentation:** 2
**Contribution:** 3
**Rating:** 2
**Confidence:** 4

**Summary:**

The authors present FLORA, a motion-flow based reward shaping method which, given a couple of demonstrations for a target task, generates a potential-based reward function for reinforcement learning of the task. FLORA requires a human to first annotate objects of relevance and the robot gripper, and then the LLM proposes candidate reward functions which are updated via a surrogate metric to encourage a well-structured reward function. Then, this reward function is deployed on the robot to update a policy. They demonstrate FLORA results on 8 tasks in Metaworld and 2 tasks on a real franka.

**Strengths:**

**Results:** Overall results are great and having real-world experiments in moderately difficult tasks (like peg insertion) is convincing. The authors demonstrate policy improvement in just 20 minutes of real-wolrld time.

**Motivation:** The problem is important and well-motivated in the context of robot policies started to actually be deployed in the real world now, and ensuring they can continue to learn using reward functions is relevant to robust deployment

**Principled:** The potential-based approach for reward shaping is principled and ensures that policies which can optimize the potential-based reward also are optimal policies for the original sparse reward task.

**Weaknesses:**

**Clarity issues: The methods section is not clearly written and should be rewritten for acceptance to ICLR**

- L225: Listing an example range of how many demonstrates constitute the “small set of successful demonstrations $\mathcal{D}_\text{demo}$” would help with providing readers context
- in Section 4.2, LLM-generated potentials are brought up at L244 but not first described. This section is quite confusing and requires the reader to skip around in the paper and figures to figure out what’s going on. There should perhaps be a better overview paragraph in this section. Specifically, upon reaching L244 for the first time, it mentions” an LLM-generated potential is prompted returns both a …” I couldn’t see how this aligns with Figure 1 showing a block of code nor corresponds to the annotation pipeline mentioned earlier.
    - This section should probably be overall rewritten significantly to first provide a more detailed overview of how all the pieces relate to each other and then describe them in detail.

**Framework:** one issue with the proposed PBRS-Milestone framework is that the robot can make negative progress towards a task (drop an object) and regress in the task threshold, yet the milestone index will stay the same. ****

**Baselines:**

- I would like to see additional baselines in MetaWorld, such as some I put in the “missing related work” below, or even one of the cited papers such as Rank2Reward which has very similar assumptions to FLORA.
- I’m unsure if the baselines are fine-tuned on the same demo data FLORA has access to (see “Questions” section), I could not find this detail in the paper.
- This is not a strict requirement, but at least 1 competitive reward model baseline in the real-world RL experiments would be nice to see.

**Missing related work:** There’s quite a few missing yet very related works that came out on arxiv/published before the ICLR 30 day concurrent work policy. Many of these are image or video input models which do not require privileged state information.

- [ReWiND](https://rewind-reward.github.io/) trains a video-language input transformer to produce dense rewards and demonstrates real-world RL fine-tuning on unseen tasks. Importantly, it demonstrates much better generalization than other similar work, perhaps making it a useful baseline due to real-world RL results along with similar assumptions to FLORA.
- [PROGRESSOR](https://arxiv.org/abs/2411.17764) demonstrates real-world reward-weighted-regression with learned reward estimator from videos
- [RoboCLIP](https://sites.google.com/view/roboclip/home): uses contrastive embedding distance as a reward function from a pre-trained video-language model
- [VICTOR](https://arxiv.org/abs/2405.16545): trains a reward model for longer-horizon tasks via subtask rewards
- [RL-VLM-F](https://rlvlmf2024.github.io/): trains a reward model using a VLM that provides binary comparison feedback instead of a human
- [FAC](https://arxiv.org/abs/2310.02635): a full pipeline that also uses a potential-based rewards, which are obtained via a value function prior. This may also serve as a useful baseline as they demonstrate real-world RL results.

**Minor issues:**

- L55: “RLdeployment”
- L55: “none of these approaches guarantee *optimal policy invariance*” — without context from first seeing section 3.2, this statement carries little meaning. Perhaps expanding on the statement will be useful.
- L67: it would be nice for the website URL link to be an actual \url{} in latex to make it clickable and highlighted
- L207: “, We” → “, we”
- L2340: “a GPT4.1” → “GPT4.1”
- L372: “hyberparameters”

**Questions:**

Do the authors believe the issue with making negative milestone progress mentioned above is a problem that can significantly impact downstream learning?

Are baselines given the same demonstrations as FLORA uses, for fine-tuning? If not, experiments should be re-run with fine-tuned VLC and LIV on those demonstrations to match the assumption of access to target-task demonstrations.

Does FLORA on the real world use SERL’s reward function too as the base? Not too many details are listed about the real-world setup in the appendix.

What is the time and compute cost of the FLORA reward potential optimization method? How long does this process take, per task? How long does human annotation take?

Does 20 minutes of real-world training mean 20 minutes of real-world deployment on-robot time? Or does it mean 20 minutes total experiment time from start to finish? Please specify.

---

### Official Review · Reviewer_KB9V · 2025-10-28

**Soundness:** 3
**Presentation:** 2
**Contribution:** 2
**Rating:** 4
**Confidence:** 4

**Summary:**

FLORA tackles the challenge for sim/real-world RL finetuning, primarily for reward shaping. Writing accurate, dense rewards is important for effective learning (compared to sparse reward),  but can be challenging for humans, because it requires expertise, and LLMs/VLMs, which can be inaccurate and require iterative improvement. FLORA addresses these specific challenges: (1) learning from high-dimensional image inputs when privileged states are not available and (2) reducing the need to validate LLM rewards/potentials before deployment. To address (1), FLORA learns from motion flows. To address (2), FLORA collects a validation dataset and uses Bayesian Optimization to find parameters to maximize the surrogate score, defined as monotonicity, subtask prediction, and PRBS-positive. Some additional fixes FLORA makes include Milestone Rewards, which is additional reward shaping provided by large positive signals after completing subtasks. FLORA shows improved results in simulated tasks and real-world tasks.

**Strengths:**

1. The paper brings together interesting ideas of reward design. There are novel ideas in the approach.
2. Individual sections / subsections of the paper are easy to understand, although I think they could be tied together more strongly.

**Weaknesses:**

1. No comparison against point-based / flow-based rewards. Comparing against these baselines can isolate whether FLORA’s improvement comes from its observation representation or the potential-based and other reward-based improvements. Example baselines: HuDOR (Guzey et. al.), FISH (Haldar et. al.), optimal transport-based rewards.
2. No other real-world RL baselines, besides SERL.
3. Writing could be improved. The method section, for example, could be better written so the reader has a clearer high-level understanding of the different challenges before delving into each one myopically.
4. Nit: Do not use acronym before defining it, no matter how simple or complex. (For example, reinforcement learning (RL) is introduced in the first line of the abstract. However, line 28, PBRS-Milestone is not defined and confusing to the reader).
5. Nit: Line 159 I believe (i) dependence on privileged states and (ii) fragility in high-dimensional domains are essentially the same challenge, as FLORA addresses it. They should not be separated into separate obstacles.
6. Nit: typo / confusing sentence in line 244

**Questions:**

1. What are the LLM prompts used?
2. What are the subtask segmentations for the sim and real tasks?
3. Is “Dense” in Figure 2 supposed to be an oracle baseline?

---

### Official Review · Reviewer_pS7n · 2025-11-01

**Soundness:** 3
**Presentation:** 2
**Contribution:** 3
**Rating:** 0
**Confidence:** 5

**Summary:**

It is a reward learning/shaping paper that uses flow-representation and LLMs. I did not review the entire paper because during the process I realized the paper is not properly anonymized. See below.

**Strengths:**

The idea made sense to me, but I didn't complete the review because the paper is not properly anonymized.

**Weaknesses:**

**The paper is not anonymized properly. The authors release the code for the algorithm. While the GitHub repo has an anonymous name, the commits are made by a person whose identity is not anonymized. Their name is clearly visible when clicked on the link in the paper. Because of this reason, I stopped reviewing the paper after page 4.**

While this is the reason for my strong reject rating, below are some feedback that I took notes of in the first 4 pages, in case they are useful to the authors for their future submissions.

1) The very first word of the abstract has a typo. It should be "reward design" instead of "rewards design."
2) The second paragraph of the Intro cites "Ma et al. 2024" and "Rocamonde et al. 2023" for VLMs providing rewards in RL. While both of them are good citations, I have two comments: (i) Rocamonde et al. paper was in ICLR 2024, not 2023, so the citation is incorrect, and (ii) Sontakke et al.'s "Roboclip: One demonstration is enough to learn robot policies" is an earlier work that uses VLMs to generate rewards, and is the first work to do so to the best of my knowledge (NeurIPS 2023). The Rocamonde et al. citation should be fixed, and the Sontakke et al. citation should be added.
3) Later in the same paragraph, the paper mentions some methods that rely on privileged state information. While it is true, there have been more recent works, such as Zhang et al.'s "ReWiND: Language-Guided Rewards Teach Robot Policies without New Demonstrations" where visual inputs are used and the reward signals are dense. This citation is missing (though I wouldn't penalize the paper for this, because this work came out only a few months before ICLR deadline).
4) Line 055 has a typo: "RLdeployment" instead of "RL deployment"
5) In Related Work, the "VLM-based Reward Design Approaches" subsection should also cite "Video2Reward: Generating Reward Function from Videos for Legged Robot Behavior Learning" by Zeng et al.
6) In the same paragraph, the paper talks about using VLMs to generate preferences over image-task pairs. While the two citations there are good, again, the original work that developed that approach is missing: "RL-VLM-F: Reinforcement Learning from Vision Language Foundation Model Feedback" by Wang et al.
7) Line 089 has a typo: "alievate" instead of "alleviate"
8) In the Background section, the problem is said to be defined as an MDP but there is no mention of a state space. Without a state definition, Markov property wouldn't hold. The paper introduces observations instead of states, so it should also be a POMDP instead of an MDP, but regardless, a state definition is still required for the Markov property to hold.

I stopped reviewing in page 4 as I realized the code repository is not anonymous.

**Questions:**

N/A

**Details Of Ethics Concerns:**

The paper is not anonymized properly. The authors release the code for the algorithm. While the GitHub repo has an anonymous name, the commits are made by a person whose identity is not anonymized. Their name is clearly visible when clicked on the link in the paper. Because of this reason, I stopped reviewing the paper after page 4.

---

### Note · Authors · 2025-11-12

I have read and agree with the venue's withdrawal policy on behalf of myself and my co-authors.